# Lyapunov Design for Robust and Efficient Robotic Reinforcement Learning

**Tyler Westenbroek**[1,*]
westenbroekt@berkeley.edu

**Fernando Castañeda**[2,*]
fcastaneda@berkeley.edu

**Ayush Agrawal**[2,*]
ayush.agrawal@berkeley.edu

**Shankar Sastry**[1]
sastry@coe.berkeley.edu

**Koushil Sreenath**[2]
koushils@berkeley.edu
[1]Department of Electrical Engineering and Computer Sciences, UC Berkeley
[2]Department of Mechanical Engineering, UC Berkeley
* Equal Contribution *

**Abstract:** Recent advances in the reinforcement learning (RL) literature have enabled roboticists to automatically train complex policies in simulated environments. However, due to the poor sample complexity of these methods, solving RL problems using real-world data remains a challenging problem. This paper introduces a novel cost-shaping method which aims to reduce the number of samples needed to learn a stabilizing controller. The method adds a term involving a Control Lyapunov Function (CLF) – an 'energy-like' function from the model-based control literature – to typical cost formulations. Theoretical results demonstrate the new costs lead to stabilizing controllers when smaller discount factors are used, which is well-known to reduce sample complexity. Moreover, the addition of the CLF term 'robustifies' the search for a stabilizing controller by ensuring that even highly sub-optimal polices will stabilize the system. We demonstrate our approach with two hardware examples where we learn stabilizing controllers for a cartpole and an A1 quadruped with only seconds and a few minutes of fine-tuning data, respectively. Furthermore, simulation benchmark studies show that obtaining stabilizing policies by optimizing our proposed costs requires orders of magnitude less data compared to standard cost designs.

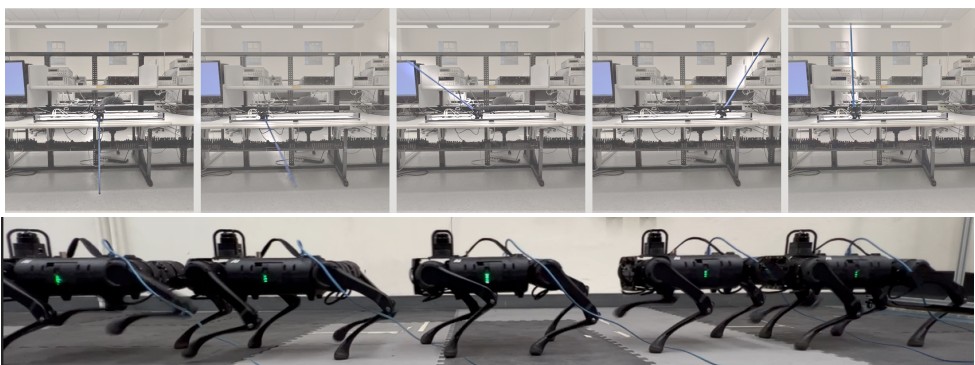

Figure 1: We learn precise stabilizing policies on hardware for the `Quanser` cartpole [1] (top) and the `Unitree A1` quadruped [2] (bottom) using only seconds and a few minutes of real-world data, respectively. A video of our experiments can be found here https://youtu.be/l7kBfitE5n8

---

*This work was supported in part by NSF Grants CMMI-1944722 and CMMI-1931853, LOGiCS (Learning-Driven Oracle-Guided Compositional Symbiotic Design of Cyber-Physical Systems), and Defense Advanced Research Projects Agency award number FA8750-20-C-0156. The work of Fernando Castañeda was partially supported through a fellowship from Fundación Rafael del Pino, Spain.

6th Conference on Robot Learning (CoRL 2022), Auckland, New Zealand.

# 1 Introduction

A key challenge in robotics is reasoning about the long-horizon behavior induced by a control policy. This is because important system properties such as stability are inherently long-horizon phenomena. In reinforcement learning (RL), the *discount factor* implicitly controls how far into the future policy optimization algorithms plan when optimizing the objective specified by the user. Standard approaches to designing objective functions for robotic RL, such as penalizing the distance to a reference trajectory, inherently require a large discount factor to learn control policies which stabilize the system [3, 4]. Unfortunately, problems with large discount factors can be extremely difficult to solve, often requiring vast data sets and careful tuning of hyper-parameters [5]. As a number of recent success stories have demonstrated [6, 7, 8, 9, 10, 11], ever-increasing computational resources can be used to solve these problems in simulation and deploy the resulting controllers directly on the real-world system. However, because it is impractical to model every detail of complex hardware platforms, achieving the best performance will require learning from real-world data.

This paper introduces a cost-shaping framework which enables users to reliably learn stabilizing control policies with small amounts of real-world data by solving problems with small discount factors. Our approach uses *Control Lyapunov Functions* (CLFs), a standard design tool from the control theory literature [12, 13, 14, 15]. CLFs are 'energy-like' functions for the system which reduce the search for a stabilizing controller to a myopic one-step criterion. In particular, any controller which decreases the energy of the CLF at each instance of time will stabilize the system. Thus, CLFs reduce the long-horizon objective of stabilizing the system to a simple one-step condition. When a CLF is available and the dynamics are known, constructive techniques from the control literature can be used to synthesize a stabilizing controller. However, when there is uncertainty in the dynamics, it is difficult to guarantee that a controller will always decrease the value of the CLF, or that we have even designed a true CLF for the system.

Our approach is to $1)$ design an approximate CLF for the real-world system using an approximate dynamics model and $2)$ modify the 'standard' choice of cost functions mentioned above by adding a term which incentivizes controllers which decrease the approximate CLF over time. This technique effectively uses the approximate CLF as supervision for reinforcement learning, enabling the user to embed known system structures into the learning process while retaining the flexibility of RL to overcome unknown dynamics. Indeed, as our analysis demonstrates, when our approach is used reinforcement learning algorithms implicitly learn to 'correct' the approximate CLF provided by the user. When the candidate CLF is close to being a true CLF for the system (in a sense we make precise below), a stabilizing controller can be efficiently learned by solving a problem with a small discount factor. Moreover, the addition of the approximate CLF 'robustifies' the search for a stabilizing controller by ensuring that even highly suboptimal policies will stabilize the system. Finally, in situations where it is too difficult to design a nominal CLF by hand, we demonstrate how one can be learned using a simulation model and the standard style of RL objective discussed above. Specifically, we use the value function learned by the RL algorithm as an approximate CLF for the real-world system. Altogether, beyond accelerating and robustifying RL, our approach also expands the applicability of CLF-based design techniques.

We apply this technique to develop data-efficient fine-tuning strategies, wherein a nominal controller developed using a simulation model is refined with small amounts of real-world data. For the A1 experiment, the nominal controller is a model-based control architecture [16], and we hand-design a CLF using a highly simplified linearized reduced-order model for the system. Even though this model is very crude, we are nonetheless able to learn a precise tracking controller for this 18 DOF system with only 5 minutes of real-world data. For the cartpole swing-up task we used the value function from a simulation-based RL problem as the candidate CLF for the real-world system, using the learning process described above. Our fine-tuning approach then learned a robust swing-up controller after observing only one 10 second trajectory from the real-world system.

## 1.1 Related Work

We outline how our approach departs from related work; Appendix A contains further discussion. **Discount Factors, Sample Complexity and Reward Shaping:** It is well-understood that the discount factor has a significant effect on the size of the data set that RL algorithms need to achieve a desired level of performance. Specifically, it has been shown in numerous contexts [17, 18, 19, 20] that smaller discount factors lead to problems which can be solved more efficiently. This has led to a number of works which explicitly treat the discount factor as a parameter which can be used to control the complexity of the problem alongside reward shaping techniques [21, 22, 5, 23, 24, 25].

Compared to these works, our primary contribution is to demonstrate how CLFs can be combined with model-free algorithms to rapidly learn stabilizing controllers for robotic systems.

**Fine-tuning with Real World Data:** Recently, there has been much interest in using RL to fine-tune policies which have been pre-trained in simulation [26, 27, 28, 29]. These methods typically optimize the same cost function with a large discount factor in both simulation and on the real robot. In contrast, using our cost reshaping techniques, we solve a different problem with a smaller discount factor on hardware which can be solved more efficiently. In Appendix D, we show that our method outperforms typical fine-tuning approaches under moderate perturbations to the dynamics model.

**Learning with Control Lyapunov Functions:** A number of recent works have also tried to overcome the reality gap using data-driven methods to improve CLF-based controllers [30, 31, 32, 33, 34, 35]. While these methods work well when a true CLF for the real-world system is available, our method is more general as we can still efficiently learn stabilizing controllers when only an approximate CLF is available by modulating the discount factor used to optimize our cost.

## 2 Background and Problem Setting

Throughout the paper we will consider deterministic discrete-time systems of the form:
$$x_{k+1} = F(x_k, u_k), \tag{1}$$
where $x_k \in \mathcal{X} \subset \mathbb{R}^n$ is the state at time $k$, $u_k \in \mathcal{U} \subset \mathcal{X}$ is the input applied to the system at that time, and $F\colon \mathcal{X} \times \mathcal{U} \to \mathbb{R}^n$ is the transition function for the system. This general nonlinear model is broad enough to cover many important continuous control tasks for robotics. We will let $\Pi$ denote the space of all control polices $\pi\colon \mathcal{X} \to \mathcal{U}$ for the system. To ease exposition, for our theoretical analysis we will focus on the case where the goal is to stabilize the system to a single point, namely the origin. Through our examples we will demonstrate how our cost-shaping technique can be leveraged to achieve more complicated tasks, and in Section 5 we outline a path for extending our theoretical results to these settings in future work.

### 2.1 Control Lyapunov Functions

Control Lyapunov Functions [12, 13, 14, 15] are 'energy-like' functions for the dynamics (1):

**Definition 1.** *We say that a positive definite function $W\colon \mathbb{R}^n \to \mathbb{R}$ is a* Control Lyapunov Function *(CLF) for* (1) *if the following condition holds for each $x \in \mathcal{X}\backslash\{0\}$:*
$$\min_{u \in \mathcal{U}} W(F(x, u)) - W(x) < 0. \tag{2}$$

The condition (2) ensures that for each $x \in \mathcal{X}$ there exists a choice of input which decreases the 'energy' $W(x)$. Any policy which satisfies the one-step condition $W(F(x, \pi(x))) - W(x) < 0$ can be guaranteed to asymptotically stabilize the system [36] (see Appendix B for background on stability theory). Given a CLF for the system, model-based methods constructively synthesize a controller which satisfies this property using either closed-form equations [13] or by solving an online (convex) optimization problem [37, 15] to satisfy (2). However, when the dynamics are unknown it is difficult to ensure that we have synthesized a 'true' CLF for the system.

**Remark 1.** *(Designing Control Lyapunov Functions) While there is no general procedure for designing CLFs by hand for general nonlinear systems, there do exist constructive procedures for designing CLFs for many important classes of robotic systems, such as manipulator arms [14] and robotic walkers [15] using structural properties of the system. Moreover, in our examples we will investigate how a CLF can be learned from a simulation model and how very coarse CLF candidates can be used to accelerate learning a stabilizing controller.*

### 2.2 Stability of Dynamic Programming and Reinforcement Learning

Here we investigate how a common class of cost functions found in the literature can be used to learn stabilizing controllers. In particular, we consider a running cost $\ell\colon \mathcal{X} \times \mathcal{U} \to \mathbb{R}$ of the form $\ell(x, u) = Q(x) + R(u)$, where $Q\colon \mathcal{X} \to \mathbb{R}$ is the state cost and $R\colon \mathcal{U} \to \mathbb{R}$ is the input cost. Both $Q$ and $R$ are assumed to be positive definite (in practice, both are usually quadratic). Given a policy $\pi \in \Pi$, discount factor $\gamma \in [0, 1]$, and initial condition $x_0 \in \mathcal{X}$, the associated long-run cost is:
$$V_\gamma^\pi(x_0) = \sum_{k=0}^{\infty} \gamma^k \ell(x_k, \pi(x_k)) \tag{3}$$
$$\text{s.t. } x_{k+1} = F(x_k, \pi(x_k)),$$

where $V_\gamma^\pi: \mathcal{X} \to \mathbb{R} \cup \{\infty\}$ is the *value function* associated to $\pi$. Small discount factors incentivize policies which greedily optimize a small number of time-steps into the future, while larger discount factors promote policies which reduce the cost in the long-run. We say that a policy $\pi_\gamma^* \in \Pi$ is *optimal* if it achieves the smallest cost from each $x \in \mathcal{X}$:

$$V_\gamma^{\pi_\gamma^*}(x) = V_\gamma^*(x) := \inf_{\pi \in \Pi} V_\gamma^\pi(x), \quad \forall x \in \mathcal{X},$$

where $V_\gamma^*: \mathcal{X} \to \mathbb{R} \cup \{\infty\}$ is the *optimal value function*. Together $V_\gamma^*$ and $\pi_\gamma^*$ capture the 'ideal' behavior induced by the cost function (3). It is well-known [17] that the optimal value function will satisfy the Bellman equation:

$$V_\gamma^*(x) = \inf_{u \in \mathcal{U}} \left[ \gamma V_\gamma^*(F(x,u)) + \ell(x,u) \right], \quad \forall x \in \mathcal{X}, \tag{4}$$

and an optimal policy $\pi_\gamma^*$ will satisfy $\pi_\gamma^*(x) \in \arg\min_{u \in \mathcal{U}} \left[ \gamma V_\gamma^*(F(x,u)) + \ell(x,u) \right], \forall x \in \mathcal{X}$. Unfortunately, it is impractical to directly search over $\Pi$ to find a policy which meets these conditions. This necessitates the use of function approximation schemes (e.g. feed-forward neural networks) to instead represent a subset of policies $\hat{\Pi} \subset \Pi$ to search over. Indeed, modern RL approaches for robotics randomly sample the space of trajectories to optimize problems of the form:

$$\inf_{\pi \in \hat{\Pi}} \mathbb{E}_{x_0 \sim X_0} \left[ V_\gamma^\pi(x_0) \right], \tag{5}$$

where $X_0$ is a distribution over initial conditions. While this approach enables these methods to optimize high-dimensional policies, they are data-hungry, can display high-variance and thus frequently return highly sub-optimal policies when data is limited. To better understand the effect that this has on the stability of learned policies, for each $\pi \in \hat{\Pi}$ and $\gamma \in [0,1]$ define the *optimality gap*:

$$\epsilon_\gamma^\pi(x) = V_\gamma^\pi(x) - V_\gamma^*(x).$$

The temporal difference equation [17] dictates that for each $x \in \mathcal{X}$ the policy satisfies:

$$V_\gamma^\pi(x) = \gamma V_\gamma^\pi(F(x, \pi(x))) + \ell(x, \pi(x)). \tag{6}$$

From these equations we can obtain:

$$V_\gamma^\pi(F(x, \pi(x))) - V_\gamma^\pi(x) = \frac{1}{\gamma} \left( -\ell(x, \pi(x)) + (1 - \gamma) V_\gamma^\pi(x) \right) \tag{7}$$

$$= \frac{1}{\gamma} \left( -\ell(x, \pi(x)) + (1 - \gamma)[V_\gamma^*(x) + \epsilon_\gamma^\pi(x)] \right) \tag{8}$$

$$\leq \frac{1}{\gamma} \left( -Q(x) + (1 - \gamma)[V_\gamma^*(x) + \epsilon_\gamma^\pi(x)] \right), \tag{9}$$

where we have first rearranged (6), then used $V_\gamma^\pi(x) = V_\gamma^*(x) + \epsilon_\gamma^\pi(x)$, and finally we have used $\ell(x, \pi(x)) \geq Q(x)$. Inequalities of this sort are the building block for proving the stability of suboptimal polices in the dynamic programming literature [4, 3].

**Remark 2.** *(Value Functions as CLFs) By inspecting the cost (3) we see that $V_\gamma^\pi$ is positive definite (since $Q$ is positive definite). Thus, if the right-hand side of (9) is negative for each $x \in \mathcal{X} \setminus \{0\}$, this inequality shows that $V_\gamma^\pi$ is a CLF for (1), and that $\pi$ is an asymptotically stabilizing control policy. In other words, $V_\gamma^\pi$ is a CLF which is implicitly learned during the training process. Indeed, many RL algorithms directly learn an estimate of the value function, a fact which we later exploit to learn a CLF for the cartpole swing up-task in Section 4 using the nominal simulation environment.*

Note that the right hand side of (9) will only be negative if $V_\gamma^*(x) + \epsilon_\gamma^\pi(x) < \frac{1}{1-\gamma} Q(x)$. Since from (3) we know that $V_\gamma^*(x) > Q(x)$ for each $x \in \mathcal{X}$, even the optimal policy (which has no optimality gap) will only be stabilizing if $\gamma$ is large enough. On the other hand, for a fixed $\gamma \in (0, 1]$, this inequality also quantifies how sub-optimal a policy can be while maintaining stability. To make these observations more quantitative we make the following assumption:

**Assumption 1.** *For each $\gamma \in [0,1]$ there exists $C_\gamma \geq 1$ such that $V_\gamma^*(x) \leq C_\gamma Q(x)$ for each $x \in \mathcal{X}$.*

Growth conditions of this form are standard in the literature on the stability of approximate dynamic programming [38, 3, 4, 39]. Note that, because the running cost $\ell$ is non-negative, we have $C_{\gamma'} \leq C_{\gamma''}$ if $\gamma' \leq \gamma''$. In particular, the constant $C_1$ upper-bounds the ratio between the one-step cost and the optimal undiscounted value function. When $C_1$ is smaller, the optimal undiscounted policy is more 'contractive' and approximate dynamic programming methods converge more rapidly to an optimal solution [38]. Thus, intuitively the constants $C_\gamma \geq 1$ will be smaller when the system is easier to stabilize. The following result is essentially a specialization of the main result from [39]:

**Proposition 1.** *Let Assumption 1 hold and let $\gamma \in [0,1]$ and $\pi \in \hat{\Pi}$ be fixed. Further assume that there exists $\delta > 0$ such that for each $x \in \mathcal{X}$ we have i) $\epsilon_\gamma^\pi(x) \leq \delta Q(x)$ and ii) $C_\gamma + \delta < \frac{1}{1-\gamma}$. Then, $\pi$ asymptotically stabilizes (1).*

*Proof.* Combining conditions $i)$ and $ii)$ with equation (9) yields:

$$V_\gamma^\pi(F(x,\pi(x))) - V_\gamma^\pi(x) \leq \frac{2}{\gamma}\big(-1 + (1-\gamma)[C_\gamma + \delta]\big)Q(x). \qquad \square$$

Thus the RHS of the preceding equation will be negative-definite if $C_\gamma + \delta < \frac{1}{1-\gamma}$, which demonstrates the desired result.

**Remark 3.** *(Stability Properties of the Cost Function) In the following section we will derive an analogous result to Proposition 1 for the novel reshaped cost function we propose below. When comparing these results we will primarily focus on the effect of the constants $C_\gamma \geq 1$ (and the equivalent constants for the new setting). The $C_\gamma$ constants can be used to bound how large of a discount factor is need to stabilize the system. In particular, Proposition 1 implies that the optimal policy will stabilize the system for each $\gamma$ which satisfies $\gamma > 1 - \frac{1}{C_\gamma}$. The $C_\gamma$ constants also characterizes how 'robust' the cost function is to suboptimal policies. In particular, for a fixed discount factor, the policy will stabilize the system if $\delta < \frac{1}{1-\gamma} - C_\gamma$. Thus smaller values of the $C_\gamma$ constants permit more suboptimal policies.*

## 3  Lyapunov Design for Infinite Horizon Reinforcement Learning

Our method uses a positive definite candidate Control Lyapunov Function $W \colon \mathbb{R}^n \to \mathbb{R}$ for the nonlinear dynamics (1), and reshapes (3) to our proposed new long horizon cost $\tilde{V}_\gamma^\pi \colon \mathcal{X} \to \mathbb{R} \cup \{\infty\}$:

$$\tilde{V}_\gamma^\pi(x_0) = \sum_{k=0}^\infty \gamma^k \bigg( [W\big(F(x_k, \pi(x_k))\big) - W(x_k)] + \ell(x_k, \pi(x_k)) \bigg) \tag{10}$$
$$\text{s.t. } x_{k+1} = F(x_k, \pi(x_k)).$$

As we shall see below, our method works best when $W$ is in fact a CLF for the system, but still provides benefits when it is only an 'approximate' CLF for the system (in a sense we will make precise later). For each $\gamma \in [0,1]$ the new optimal value function is given by:

$$\tilde{V}_\gamma^*(x) = \inf_{\pi \in \Pi} \tilde{V}_\gamma^\pi(x). \tag{11}$$

The new cost (10) includes the amount that $W$ changes at each time step, and thus encourages choices of inputs which decrease $W$ over time. In this case, the Bellman equation [17] dictates:

$$\tilde{V}_\gamma^*(x) = \inf_{u \in \mathcal{U}} \big[\gamma \tilde{V}_\gamma^*(F(x,u)) + \Delta W(x,u) + \ell(x,u)\big], \quad \forall x \in \mathcal{X}, \tag{12}$$

where $\Delta W(x,u) := W(F(x,u)) - W(x)$. To gain some intuition for the approach let us consider the two extremes where $\gamma = 0$ and $\gamma = 1$. In the case where $\gamma = 1$, by inspection we see that $\tilde{V}_1^* = V_1^* - W$ solves the Bellman equation. Plugging in this solution demonstrates that any optimal policy $\tilde{\pi}_1^*$ must satisfy $\tilde{\pi}_1^*(x) \in \arg\min_{u \in \mathcal{U}}[V_1^*(F(x,u)) + \ell(x,u)]$. This is precisely the optimality condition for the original cost (3) when $\gamma = 1$, and thus the set of optimal policies for the two problems coincide. Thus, in this case, by embedding the CLF in the cost we are effectively using $W$ as a warm-start initial guess for the optimal value function. In the other extreme where $\gamma = 0$, from (12) we see that an optimal policy must satisfy $\tilde{\pi}_0^*(x) \in \arg\min_{u \in \mathcal{U}}\big[\Delta W(x,u) + \ell(x,u)\big]$. Thus, when $\gamma = 0$ the optimal policy attempts to greedily decrease the value of the candidate CLF and the one-step cost on the input. As we shall see below, when intermediate discount factors are used, optimal policies may instead decrease the value of $W$ over the course of several steps.

Using the new cost function (10), each policy must satisfy the new difference equation:

$$\tilde{V}_\gamma^\pi(x) = \gamma \tilde{V}_\gamma^\pi\big(F(x,\pi(x))\big) + W\big(F(x,\pi(x))\big) - W(x) + \ell(x,\pi(x)). \tag{13}$$

In our stability analysis, we will use the following composite function as a candidate CLF for (1):

$$\tilde{\mathcal{V}}_\gamma^\pi(x) = W(x) + \gamma \tilde{V}_\gamma^\pi(x). \tag{14}$$

We provide an interpretation of this curious candidate CLF in Remark 4 below, but first perform an initial analysis similar to the one presented in the previous section. Defining for each $\pi \in \hat{\Pi}$, $\gamma \in [0, 1]$ and $x \in \mathcal{X}$ the new optimality gap:

$$\tilde{\epsilon}_\gamma^\pi(x) = \tilde{V}_\gamma^*(x) - \tilde{V}_\gamma^\pi(x), \tag{15}$$

and following steps analogous to those taken in (7)-(9), we can obtain the following:

$$\tilde{\boldsymbol{\mathcal{V}}}_\gamma^\pi\big(F(x, \pi(x))\big) - \tilde{\boldsymbol{\mathcal{V}}}_\gamma^\pi(x) = -\ell(x, \pi(x)) + (1 - \gamma)\tilde{V}_\gamma^\pi(x) \tag{16}$$

$$= -\ell(x, \pi(x)) + (1 - \gamma)\big[\tilde{V}_\gamma^*(x) + \tilde{\epsilon}_\gamma^\pi(x)\big] \tag{17}$$

$$\leq -Q(x) + (1 - \gamma)\big[\tilde{V}_\gamma^*(x) + \tilde{\epsilon}_\gamma^\pi(x)\big]. \tag{18}$$

Similar to the analysis in the previous section, we will aim to understand when the right-hand side of (18) is negative, as this will characterize when $\pi$ stabilizes the system. One key difference between the inequalities (9) and (18) is that, while the original value function $V_\gamma^*$ is necessarily positive definite, $\tilde{V}_\gamma^*$ can actually take on negative values since the addition of the CLF term allows the new running cost in (10) to be negative. As we shall see, this forms the basis for the stability and robustness properties our cost formulation enjoys when $W$ is designed properly.

**Remark 4.** *(Learning Corrections to W) When the right hand side of (18) is negative for each $x \in \mathcal{X} \setminus \{0\}$, inequality (18) demonstrates that $\tilde{\boldsymbol{\mathcal{V}}}_\gamma^\pi$ is in fact a CLF for (1) and that $\pi$ stabilizes the system (see Theorem 1). We can think of $W$ as an 'initial guess' for a CLF for the system, while $\gamma \tilde{V}_\gamma^\pi$ is a 'correction' to $W$ that is implicitly made by a learned policy $\pi$. Roughly speaking, the larger the discount factor, the larger this correction. Thus, the user can trade-off how much the learned policy is able to correct the candidate CLF $W$ against the additional complexity of solving a problem with a higher discount factor, depending on how 'good' they believe the CLF candidate to be.*

We first state a general stability result for suboptimal policies associated to the new cost, and then discuss how the choice of $W$ affects the stability of suboptimal control policies:

**Assumption 2.** *For each $\gamma \in [0, 1]$ there exists $\tilde{C}_\gamma \in \mathbb{R}$ such that $\tilde{V}_\gamma^*(x) \leq \tilde{C}_\gamma Q(x)$ for each $x \in \mathcal{X}$.*

Because the reshaped one-step cost $W(F(x, u)) - W(x) + \ell(x, u)$ can take on negative values, so can the $\tilde{C}_\gamma$ constants. Moreover, in this case it is possibe to have $\tilde{C}_{\gamma'} \geq \tilde{C}_{\gamma''}$ when $\gamma' \leq \gamma''$. This is because when larger discount factors are used, the optimal policy can benefit from decreasing $W$ further into the future. The following stability result is analogous to Proposition 1:

**Theorem 1.** *Let Assumption 2 hold and let $\gamma \in [0, 1]$ and $\pi \in \hat{\Pi}$ be fixed. Further assume that there exists $\tilde{\delta} > 0$ such that for each $x \in \mathcal{X}$ we have i) $\tilde{\epsilon}_\gamma^\pi(x) \leq \delta Q(x)$ and ii) $\tilde{C}_\gamma + \tilde{\delta} < \frac{1}{1-\gamma}$. Then, $\pi$ asymptotically stabilizes (1).*

The proof is conceptually similar to the proof of Proposition 1; we delegate the proof to Appendix C for brevity. Indeed, note that the conditions for stability under the new cost are essentially identical to those for the previous cost in Proposition 1.

As alluded to in Remark 3, we will primarily focus on comparing how large the constants $C_\gamma \geq 1$ and $\tilde{C}_\gamma \in \mathbb{R}$ are for the two problems, as they control the discount factor required to learn a stabilizing policy and also the 'robustness' of the cost to suboptimal controllers. We provide two characterizations which ensure that $\tilde{C}_\gamma < C_\gamma$. The first condition is taken from the model-predictive control literature [40, 41], where CLFs are used as terminal costs for finite-horizon prediction problems. Proof of the following result can be found in Appendix C:

**Lemma 1.** *Suppose that for each $x \in \mathcal{X}$ the following condition holds:*

$$\inf_{u \in U} W(F(x, u)) - W(x) + \ell(x, u) \leq 0. \tag{19}$$

*Then Assumption 2 is satisfied with constant $\tilde{C}_\gamma \leq 0$.*

The hypothesis of Lemma 1 implies that $i)$ $W$ is a true CLF for the system and $ii)$ $W$ dominates the running cost $\ell$, in the sense that $W$ can be decreased more rapidly than $\ell$ accumulates. Effectively, this condition implies that it is advantageous for polices to myopically decrease $W$ at each time step. Consequently, when this condition holds optimal polcies associated to the reshaped costs (10) will stabilize the system for any choice of discount factor.

The following definition generalizes this condition to cases where $W$ may not be a true CLF for the system but can be decreased over several time-steps:

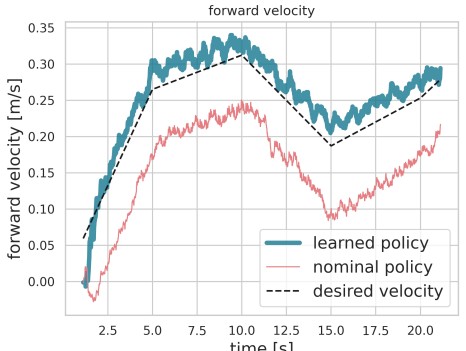
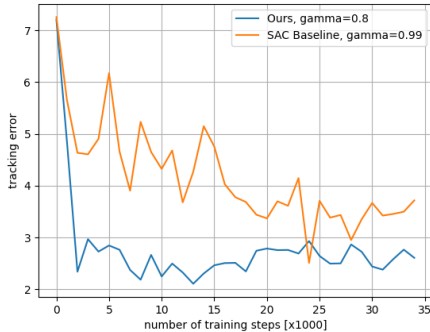

Figure 2: (Left) Plot illustrating improved velocity tracking of the learned policy (in dark green) compared to the nominal locomotion controller (in pink) to track a desired velocity profile (in dashed black line) using our proposed method on the `Unitree A1` robot hardware. (Right) Plot from the simulated benchmark study illustrating cumulative velocity tracking error (lower is better) over 10s rollouts at different stages of the training. In orange, we show the results of fine-tuning using SAC with a standard RL cost. In blue, we fine-tune using SAC with our reward reshaping method, with a candidate CLF designed on a nominal linearized model of the robot. In both cases, we plot the results using the discount factor that achieved the best performance.

**Definition 2.** *We say that the candidate CLF $W$ $\bar{\gamma}$-dominates the running cost $\ell$ if for each discount factor $\bar{\gamma} \leq \gamma \leq 1$ and $x \in \mathcal{X}$ we have $\tilde{V}_\gamma^*(x) \leq V_\gamma^*(x)$.*

The condition in (2) effectively provides a way of characterizing how 'close' $W$ is to being a true CLF for the real-world system. In particular, the larger $\bar{\gamma}$ the further into the future RL algorithms must look to see the benefits of decreasing $W$. Our previous discussion, which showed that $\tilde{V}_1^* = V_1^* - W$, demonstrates that every candidate CLF 1-dominates the cost. Moreover, clearly $W$ can only 0-dominate the original cost if it is a CLF for the system. While this condition is more difficult to verify for intermediate values of $\bar{\gamma}$, it provides qualitative insight into how even approximate CLFs for the system can still make it easier to obtain stabilizing controllers.

**Remark 5.** *(Robustness of reshaped cost) When the condition of Lemma 1 is satisfied we will have $\tilde{C}_\gamma \leq 0 < C_\gamma$, implying the new cost enjoys the desirable robustness properties discussed above. When $W$ satisfies the 'approximate CLF' condition in Definition (2), it will only enjoy these benefits when the discount factor is large enough. We leave it as a matter for future work to provide quantitative estimates for the $\tilde{C}_\gamma$ constants in these regimes, and to provide sufficient conditions which ensure $W$ $\bar{\gamma}$-dominates the running cost.*

## 4 Examples and Practical Implementations

We summarize the main results for each of our examples, but leave most details and plots to Appendix D. In every experiment we report, the soft actor-critic algorithm (SAC) [42] is used as the learning algorithm to optimize the various reward structures we investigate.

**Velocity Tracking for A1 Quadruped:** We apply our approach to train a neural network controller which augments and improves a nominal model-based controller [16] for a quadruped robot using real-world data. As illustrated by the pink curve in Fig. 2 (left), the nominal controller fails to accurately track desired velocities specified by the user. We design a CLF around the desired gait using a linearized reduced-order model for the system. We then collect rollouts of $10s$ on the robot hardware with randomly chosen desired velocity profiles, and solve an RL problem using our cost and a discount factor $\gamma = 0$. Our approach is able to learn a policy which significantly improves the tracking performance of the nominal controller within 5 minutes (30 episodes) of hardware data, as shown in Fig. 2 (left). A video of these results can be found in https://youtu.be/ l7kBfitE5n8, and more details are provided in Appendix D. Furthermore, in Fig. 2 (right) we benchmark our approach in simulation against an RL agent trained with a 'standard' cost which penalizes the squared error with respect to the desired velocity. As this figure demonstrates, our method is able to rapidly decrease the average tracking error in only around 2 thousand steps from the environment. In contrast, the benchmark approach is only able to reach this level of performance for the first time after around 24 thousand steps.

**A1 Quadruped Walking with an Unknown Load:** We attach an un-modeled load to the A1 quadruped, that is equivalent to one-third the mass of the robot. Fine-tuning on hardware the same base controller from the previous set-up where the CLF is designed to stabilize to the target gait, our approach is able to significantly decrease the tracking error to about one-third its nominal value with only one minute of data collected on the robot hardware as illustrated in Fig. 3 in Appendix D. Additionally, in Appendix D, we run a simulated benchmark comparison and verify that our method clearly out-performs the 'standard' cost baseline for this task.

**Fine-tuning a Learned Policy for Cartpole Swing-Up:** We fine-tune a swing-up controller for the Quanser cartpole system [1] using real-world data and an initial policy which was pre-trained in simulation but that does not translate well to the real system. Due to the underactuated nature of the system, synthesizing a CLF by hand is challenging. Thus, as alluded to previously, we use a 'typical' cost function of the form (3) and a discount factor of $\gamma = 0.999$ to learn a stabilizing neural network policy $\pi_\phi$ for a simulation model of the system. Given the discussion in Remark 2, we use the value function $V_\theta$ associated with the simulation-based policy as the candidate CLF ($W = V_\theta$) for our reward reshaping formulation (10). When improving the simulation-based policy $\pi_\phi$ with real-world data, we keep the parameters of this network fixed and learn an additional smaller policy $\pi_\psi$ (so that the overall control action is produced by $\pi_\phi + \pi_\psi$) using our proposed CLF-based cost formulation. We solve the reshaped problem with a discount factor $\gamma = 0$ and collect rollouts of $10s$ on hardware. Our CLF-based fine-tuning approach is able to successfully complete the swing-up task after collecting data from just one rollout. After collecting data from an additional rollout, the controller is reliable and robust enough to recover from several pushes. A video of these experiments can be found in https://youtu.be/l7kBfitE5n8, and more details and plots of the results are provided in Appendix D. Furthermore, in Appendix D we provide a simulation study comparing a standard fine-tuning approach to our method, showing that our approach is able to more rapidly learn a reliable swing-up policy than the baseline and also achieves a higher reward.

**Fine-tuning a Bipedal Walking Controller in Simulation:** We also apply our design methodology to fine-tune a model-based walking controller [15] for a bipedal robot with large amounts of dynamics uncertainty. Model uncertainty is introduced by doubling the mass of each link of the robot. The nominal controller fails to stabilize the gait and falls within a few steps. To apply our method, we design a CLF around the target gait as in [15] to be used in our reward formulation. As a benchmark comparison, we also train policies with a reward which penalizes the distance to the target motion (no CLF term), as is most commonly done in RL approaches for bipedal locomotion which use target gaits in the reward [10]. Our approach is able to significantly reduce the average tracking error per episode after only 40000 steps of the environment (corresponding to 40 seconds of data), while the baseline does not reach a similar level of performance even after 1.2 million steps, as illustrated in Fig. 7 of Appendix D.

**Inverted Pendulum with Input Constraints:** Our final example demonstrates the utility of our method even when $W$ is a crude guess for a CLF for the system, through the use of moderate discount factors. We illustrate this for a simple inverted pendulum simulator by varying the magnitude of the input constraints for the system. We use the procedure from [15] to design a candidate CLF for the system. Like many CLF design techniques, this approach assumes there are no input constraints and encourages the pendulum to swing directly up. As the input constraints are tightened, $W$ becomes a poorer candidate CLF, as there is not enough actuation authority to decrease $W$ at each time step. Even in this case, in line with the discussion of Remark 5, if a proper discount factor is used, the addition of the candidate CLF in the reward enables our method to rapidly learn a stabilizing controller for each setting of the input bound. These results are presented in Appendix D.

## 5    Discussion and Limitations

As we have mentioned previously, our approach has several limitations. The cost-shaping technique we introduce in Section 3 only provides benefits when $W$ is in-fact a reasonable guess for a CLF for the true system. This requires that the user has a dynamics model which captures the primary features of the environments which affect the structure of CLFs for the system. While the cart-pole simulations we provide in the Appendix D provide some intuition for when this will be the case, further research is needed to better understand in what scenarios we can see significant benefits from our method. Nonetheless, our two hardware experiments provide encouraging initial results which indicate that our method can rapidly learn stabilizing controllers using CLFs which are constructed using a nominal dynamics model. More broadly, there are many exciting avenues for further incorporating Lyapunov design techniques with RL, especially offline learning [43].

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
