# OpenReview forum: "Lyapunov Design for Robust and Efficient Robotic Reinforcement Learning"
_robot-learning.org/CoRL/2022/Conference — CoRL 2022 Poster_

### Official Review · Reviewer_9gjg · 2022-07-27

**Originality:** Good
**Technical Quality:** Good
**Clarity Of Presentation:** Good
**Impact:** 2

**Recommendation:**

Weak Accept: I recommend accepting the paper, but will not argue for my recommendation if the majority of other reviewers have a different opinion.

**Summary:**

This  paper proposed a novel way of utilizing Lyapunov functions as the reward function to accelerate the RL process with small discount parameter. Both theorical analysis and real hardware experiment were provided.

**Issues:**

Please address my comments above.

**Quality Of The Limitations Section:**

Limitations are not well addressed

**Reviewer Expertise:**

4: The reviewer is confident but not absolutely certain that the evaluation is correct

**Robotics Focus:**

Relevant but unlikely to deploy to hardware in near future

**Strengths And Weaknesses:**

Strengths
1. The toptic is very interesting
2. The theorical analysis could support the proposed method

Weaknesses
The main concern of the reviewer is about the experimental results. At the begining of the paper, the experiment looks very cool wicth includes both pendulum and quadruped. On the other hand, the  actual expriment design is not that attractive:

The pendulum task is relative easy for SAC. In this paper, the value function of SAC learned in simulation was selected as the CLF, one part of the reward function to help the policy based RL to quickly learn the real task with zero discount parameter. The policy is based on the SAC neural network which has learned relatively good feature to this task; The reward is also based on the value function which is for modeling the long-term reward with the previous discount parameter. The quadruped experiment is over simplified, especially when the CLF is already given.

Overall,  It is difficult to judge the practical value of CLF without addressing how hard the current tasks are for existing RL approach.

**Summary Of Recommendation:**

Weak rejection for the limition in the current experiment.

---

> ### Author Response · Authors · 2022-08-24
> **Initial Response to Comments**
>
> $\textbf{Update: since our initial response we have added additional experiments and baseline comparisons to the draft demonstrating ourmethod}$
> $\textbf{consistently out-performs base-line cost formulations. These include new experiments with the A1 and a biped simulation model.}$
>
>
> We agree with the reviewer that the case for the paper will be made stronger with more convincing empirical demonstrations. In our response to the AC above we have detailed additional experiments results we have obtained and are working on (including bipedal walking in simulation and an additional quadruped task). Below, we would like to clarify our argument for why the results in the initial submission are compelling:
>
> We actually completely agree with the reviewer that the problems we are solving on hardware are `simplified’. However, this is actually the goal of our approach: to use Lyapunov functions to specify RL objectives which are easy for standard algorithms such as SAC to optimize but still encode complicated desired behaviors (swing-up and walking with an 18DOF robot). This enables us to generate these complex behaviors on hardware with small amounts of data (when compared to ‘typical’ cost functions).
>
> We address your specific comments on the two experiments below:
>
> **Cartpole Pendulum**: In the appendix of the initial submission we directly compared the sample complexity of our approach to approaches which use 'standard cost' functions. For the current draft we have moved this comparison to the main paper to more clearly highlight how our approach significantly decreases the amount of data needed to learn a stabilizing controller compared to standard fine-tuning approaches using SAC. In particular, as the reviewer points out, this example demonstrates how our approach can be used to learn good features for the task using a simulator, and then solve an `easy’ problem with hardware data to achieve the desired task. As our benchmark comparison demonstrates, our approach significantly decreases the sample complexity of the task compared to the baseline.
>
> **Quadruped Experiments**: For the quadruped, we feel that it is actually a strength of our approach that the CLF is given and does not need to be learned from scratch (like the cartpole). In particular, CLFs are a common tool used to design stable walking gaits [39], and a framework domain experts are comfortable working with. Thus, we argue that this example illustrates how our framework provides a natural framework for roboticists to embed domain knowledge into reinforcement learning reward functions to make the learning problem easier to solve on hardware. As our benchmark comparison for this task demonstrates, our method significantly reduces the number of samples needed to generate an accurate gait.
>
> Please let us know if we have addressed your concerns, and please let us know if there are any additional experiments you believe we can run to make the evaluation more compelling.

---

### Official Review · Reviewer_eFRG · 2022-08-01

**Originality:** Fair
**Technical Quality:** Good
**Clarity Of Presentation:** Very Good
**Impact:** 3

**Recommendation:**

Weak Accept: I recommend accepting the paper, but will not argue for my recommendation if the majority of other reviewers have a different opinion.

**Summary:**

The paper considers the problem of the poor sample complexity of RL. To address the problem, the paper introduces a cost-shaping method, which adds a control Lyapunov function (CLF) to the cost formulation. The paper theoretically proved that the proposed cost formulation can lead to a stable controller when small discount factors are used, and therefore reduces the sample complexity. The proposed method is demonstrated on a cartpole and an A1 quadruped.

**Issues:**

1. The comparsion of the proposed method and the aforementioned two works is missing. Could you state the new contribution of the paper and the key difference of the reward design technique? Could you compare the proposed work with the aforementioned two works in experiments if the problem settings are the same?

2. The problem statement is not very clear to me. Do you want to maximize the cummulative reward (standard RL setting), or just to make the learned controller stable? In the former case, will the added term in the value function make the cummulative reward higher? This is not supported by the experimental results. In the latter case, what's the function of the reward function if you have a CLF approximation?

3. One of the advantanges of the proposed algorithm is the increased sample efficiency. However, this advantage is only proved by a simple inverted pendulum system. For the carpole and the Unitree A1 robot, the paper only provides the conparsion of the fine-tuned policy and the nominal policy, which cannot support the authors' claim about the sample efficiency. In addition, the conparsion of the fine-tuned policy and the nominal policy seems a bit trivial since the fine-tuned policy is supposed to be better. What if we only fine-tune the policy using the original reward function in the real world?

4. In Figure 4, the control limits, the discount factor, and the existance of the CLF vary simultaneously. This is not sufficient to show the efficacy of the proposed method as we cannot observe which specific variable controls the change of the performance. The authors may want to provide more ablation studies to settle this issue.

5. The ablation of the approximation of the CLF is not provided. Will the quality of the approximation affect the learned controller a lot?

6. Could you provide the cummulative reward of the trajectories w.r.t. the training epochs? Since RL is proposed to maximize the cummulative reward, can the proposed method do better?

7. Can you provide some inituitions behind the assumptions propsed in the paper?


**Quality Of The Limitations Section:**

Limitations are addressed clearly

**Reviewer Expertise:**

4: The reviewer is confident but not absolutely certain that the evaluation is correct

**Robotics Focus:**

Sufficient demonstration on hardware

**Strengths And Weaknesses:**


The problem that the paper considers is important. The proposed method is technically sound, and the experiments demonstrate that the proposed method can indeed stablize the system with higher sample efficiency in real-world environments. The writing of the paper is good.

However, the most important weakness of this paper is that adding a CLF term in the cost function of RL is not a novel idea. Several previous works [1,2] use the same idea to address the stability issues in RL. Moreover, [2] does not need a given CLF approximation, but learns the CLF with the RL process directly. Therefore, the authors may want to discuss their contributions more clearly compared with [1] and [2]. Other weaknesses include the unclear problem statement, the missing comparisons with baselines, the missing ablation studies, and the missing details of the experiments.

[1] Minghao Han, Lixian Zhang, Jun Wang, and Wei Pan. Actor-critic reinforcement learning for control with stability guarantee. IEEE Robotics and Automation Letters, 5(4), pages 6217-6224, 2020.

[2] Ya-Chien Chang and Sicun Gao. Stabilizing neural control using self-learned almost lyapunov critics. In 2021 IEEE International Conference on Robotics and Automation (ICRA), pages 1803–1809. IEEE, 2021.

**Summary Of Recommendation:**

Although the paper considers an important problem, the conparison with the previous work is not enough. Two aforementioned papers seem to solve the same problem under milder assumptions. In addition, there are issues with the problem statement, and the experimental results cannot fully support the claims that the paper makes. Therefore, I give this paper a weak reject.

-----------Post discussion Note-----------
I would like to thank the authors for providing additional explanations, experiments, and the revision. The additional comparison with the two works I brought up has addressed my concerns about the novelty of the work. I agree that this work can be complementary to the two works I mentioned. In addition, I found that the additional explanations and details of the experiments are clear and can address my concerns. It could be better if the authors provide more details about how to choose the best discount factors in practice and how the performance of the proposed algorithm changes with different discount factors. I would like to change my score to a weak accept.

---

> ### Author Response · Authors · 2022-08-24
> **Initial response to comments**
>
> $\textbf{Update: since our initial response we have added additional experiments and baseline comparisons to the draft demonstrating our}$
> $\textbf{consistently out-performs base-line cost formulations. These include new experiments with the A1 and a biped simulation model.}$
>
> We thank the reviewer for pointing us to the two pieces of related work and for pointing out parts of the initial submission that were unclear. We would value the reviewers feedback on the new draft we have uploaded to see if we have improved clarity. We have included a detailed description of our changes to experiments in our response to the AC above, including additional comparisons between our method and baselines. We first address your major comments:
>
> **Major Comments:**
>
> - **Problem statement**: Our central claim is that our approach reduces the sample complexity of learning a controller which achieves the stabilization task. Thus, our goal is not to maximize the discounted sum of rewards for the original cost function (although our experiments indicate that this is a secondary benefit of our approach). We have included comments in the new draft to make this clear (see remark 6).
>
> - **Comparison to prior work**: In both provided references, learning the candidate CLF (typically from long-horizon rewards) is a crucial part of the proposed novel actor-critic algorithms. By explicitly penalizing the Lyapunov constraint, these approaches learn better CLF candidates and stabilizing controllers than standard actor-critic approaches. Thus, we argue that these references provide algorithmic improvements (over standard actor critic approaches). In contrast, our approach is agnostic to the particular RL algorithm that is used. Instead, we propose a two-step design process where we 1) assume  a reasonable CLF candidate has been constructed offline and 2) greedily optimize our new reward with a small discount factor to rapidly learn a stabilizing policy. Thus, our approach is complementary to the references, as they can be used to learn a high-quality CLF in simulation before our fine-tuning approach is applied.
>
>
>
>
> **Responses to *Issues*:**
>
> 1) As discussed above, we are working in different settings than the actor-critic methods you referenced. We are, however, looking into whether we can use these approaches as the base algorithm for our design procedure.
>
> 2) Addressed above.
>
> 3) See comments to AC for additional baselines we provide.
>
> 4) Thank you for bringing this point up – we did not realize our description of these experiments was unclear in the initial submission. As we describe more clearly in the new draft, our goal is to see how our method performs when the CLF candidate is ‘poor’. We vary the input limits to control the quality of the CLF (smaller limits means a worse CLF). Then, for each of these scenarios we sweep across discount factors to find which discount factor leads to the best performance for each of the cost formulations (with and without CLF), and plot the performance of the resulting agent in Figure 9. In summary: we benchmark our approach against the original cost function in three scenarios (3 different input constraints) and ablate the discount factor to see which cost function is superior. We feel that figure 9 is the most concise way to demonstrate that our method outperforms the baseline in all 3 scenarios. However, we would value your feedback after reading our clarifications.
>
> 5) As in the previous comment, Figure 9 shows that our method outperforms the baseline even when the CLF is quite 'bad'.
>
> 6) We have plotted the cumulative reward per-epoch for our benchmark comparisons. As these plots show, even though our primary goal is not to increase the cumulative reward of the original cost, our method still beats the baseline on these comparisons.
>
> 7) Our two main assumptions are Assumptions 1 and 3 about the growth rate of the value functions. In our previous draft, we commented after Assumption 1 that these assumptions are related to how contractive the optimal policy is (which is also related to how quickly dynamic programming methods converge to the optimal solution [45]). We have tried to make this more explicit in the new draft. Assumptions 2 & 4 are used primarily to fix notation describing how sub-optimal policies are. Finally, the condition in equation (22) (in Lemma 1) characterizes how quickly a greedy policy will decrease the CLF at each time-step.

---

### Official Review · Reviewer_ACEu · 2022-08-05

**Originality:** Good
**Technical Quality:** Very Good
**Clarity Of Presentation:** Very Good
**Impact:** 2

**Recommendation:**

Weak Accept: I recommend accepting the paper, but will not argue for my recommendation if the majority of other reviewers have a different opinion.

**Summary:**

The authors address the problem of efficiently learning and fine-tuning stabilising controllers on hardware by introducing a novel cost-shaping method based on Control Lyapunov Functions (CLF). CLFs provide some guarantees about the stability of controllers and an objective that can more greedily be optimised. The latter is useful for data efficiency, as it allows to learn with a smaller discount. The authors provide some bounds on the range of stable discounts given the sub-optimality of a policy. Experiments include fine-tuning a cartpole controller on hardware in seconds, a velocity-tracking controller on a quadruped in minutes, and an inverted pendulum in simulation, the latter for which shows the flexibility and robustness of the approach to approximate CLFs.

**Issues:**

- I would recommend to include the hardware fine-tuning baseline in the main paper
- There are "Remark", "Assumption", "Proposition", "Definition", "Theorem" and "Lemma" counters, perhaps these can be omitted to improve readability?
- Minor typos:
  - l120: R is the input [cost]
  - l131: of the from -> of the form
  - l140: the inequality needs to be flipped?
  - l176: + C -> - C
  - l204: missing \gamma

**Quality Of The Limitations Section:**

Additional details required

**Reviewer Expertise:**

3: The reviewer is fairly confident that the evaluation is correct

**Robotics Focus:**

Sufficient demonstration on hardware

**Strengths And Weaknesses:**

Strengths:
- Data efficiency in fine-tuning is a critical topic for robot learning research
- Mixing in ideas from control theory comes with convenient benefits, such as theoretical guarantees and bounds that can help tune parameters (e.g. discount)
- Hardware experiments on both a cartpole and a quadruped
- Flexibility and robustness to choice of CLF, incl. using only a locally-valid approximation and a value function trained purely in sim
- The latter especially may make this approach fairly easily widely applicable
- Paper reads well and provides sufficient theoretical backing

Weaknesses:
- Quite strong assumptions about the nature of the control problem to be solved (stabilising, deterministic, fully-observed, specific shape of cost), though this is pretty common in more theoretically-based work
- The fine-tuning results for the hardware experiments should be included in the main paper

**Summary Of Recommendation:**

Overall I'm leaning towards accepting this submission. It provides a cleanly-presented, theoretically-founded approach for improving data efficiency during fine-tuning, and the results seem to prove the effectiveness. I'm reluctant to make it a strong accept due to the underlying assumptions of the approach and how well those will hold for modern robotics problems, e.g. learning to manipulate from pixels.

---

> ### Author Response · Authors · 2022-08-24
> **Response to initial comments**
>
> $\textbf{Update: since our initial response we have included new additional experiments and benchmarks.  Please see the response to the AC for details }$
>
> We thank the reviewer for their detailed reading of the paper and the constructive comments. We agree with the limitations the reviewer identified in the present work, have addressed these points more thoroughly in the expanded discussion section we have uploaded, and are actively planning to address these shortcomings in future work. We briefly summarize our rationale for not addressing these topics in the current paper:
>
> - As the reviewer identified, there are more complicated forms of cost functions which are used in practice. In short, we actively chose to focus on a simple theoretical set-up for this paper to streamline the presentation of our main ideas regarding sample complexity. In particular, our analysis is built on insights from the MPC literature (see Appendix A) where more complicated set-ups have also been considered (e.g. tracking vs. stabilization, partially observable, stochastic, more general non-positive definite costs).  As we outline in our new draft, we believe there is a natural pathway to translate these results into our theoretical framework. Thus, our goal with this submission was to provide initial theoretical justification for our approach with empirical backing.
>
>
> - While we have focused primarily on motor control, we agree that it is not immediately clear how to reconcile our approach with other learning paradigms for robotics (e.g. learning from pixels). Indeed, there are still many unanswered questions about how control theoretic concepts such as Lyapunov functions appear in these settings. However, consider for example the work “The surprising effectiveness of linear models for visual foresight in object pile manipulation” (Suh and Tedrake 2020), which investigated how to define distance and Lyapunov functions for vision-based manipulation tasks. We believe our formulation is compatible with such approaches and can naturally leverage future insights into the stability properties of such systems. Thus, once again, our goal with this paper is to encourage research in directions such as this.
>
> Please let us know if we have addressed your concerns regarding the broader impact of our method and theory.

---

### Official Review · Reviewer_XRvu · 2022-08-05

**Originality:** Fair
**Technical Quality:** Good
**Clarity Of Presentation:** Very Good
**Impact:** 2

**Recommendation:**

Weak Accept: I recommend accepting the paper, but will not argue for my recommendation if the majority of other reviewers have a different opinion.

**Summary:**

This paper investigates sample-efficient fine-tuning of controllers by adding an approximate control Lyapunov function to the conventional cost employed for policy learning. This addition enables the usage of smaller discount factors during adaptation. The approach is demonstrated for sim2real transfer on a pendulum swing-up task, as well as reducing tracking error of a model-based controller on a quadrupedal robot.

**Issues:**

- Expand discussion of assumptions and limitation
- Details on value function quality for the pendulum SAC experiment, how is down-stream performance affected over different training snapshots of the first stage
- Expand discussion of quadruped tracking

**Quality Of The Limitations Section:**

Additional details required

**Reviewer Expertise:**

3: The reviewer is fairly confident that the evaluation is correct

**Robotics Focus:**

Sufficient demonstration on hardware

**Strengths And Weaknesses:**

**Strengths:**

The paper focuses on an important topic as training policies for robotic hardware (from scratch) is generally expensive, and being able to efficiently fine-tune a given controller to the specific hardware setup can be crucial. Overall the paper is well-written and formatted, clearly states its assumptions and does not overstate its contribution. The authors provide a diverse set of experiments including fine-tuning with a learned policy/value function as well as adapting based on a model-based controller. The hardware demonstrations further underline real-world applicability.

**Weaknesses:**

The paper would profit from elaborating on failure modes and limitations. For example, it would be helpful to discuss the assumption of positive definite cost matrices in the context of general RL problems. It would furthermore be interesting to ablate on the quality of the learned value function (pendulum) to see how well-suited the value function is at different stages of SAC training (or even alternative algorithms). The results in Figure 3 show that the velocity tracking error has been reduced, in fact it now leads the target instead of lagging (elaborate on why?). The resulting controller now appears more shaky than before, which may be due to visualization. It would have been interesting to see a video of the resulting gait to better assess qualitative performance, but a discussion of these phenomena would be good. There are also several instances where the reader is referred to prior work (e.g. construction of CLF), while inclusion in the appendix would make the paper more self-contained.

**Minor:**
- Line 55: hand-desired --> hand-designed
- Figure 2: increasing legend and axis label size for better visibility
- Figure 4: could be helpful to provide dotted red lines to highlight x & y intercepts of black dots;
- Figure 4: floating line above right panel

**Summary Of Recommendation:**

The paper aims to address the important topic of sample-efficient fine-tuning particularly in light of hardware transfer, which is valuable for the robotics community. The exposition is generally well-written, while it would benefit from some expansion of the limitations section (see above). Overall, the paper illustrates the effectiveness of the proposed formulation sufficiently well in the context of the experiments, although additional experimental evaluation would help strengthen the paper.

------- Post rebuttal -------

The authors have addressed the majority of my comments justifying an increase to weak accept.

---

> ### Author Response · Authors · 2022-08-24
> **Initial response to comments**
>
> We thank the reviewer for their detailed comments and agree with the shortcomings in discussion in the initial draft. We have rearranged the current draft to allow for more thorough evaluations in the main paper. As discussed in our response to the meta-review, we are generally providing more experiments to bolster our claims. We are addressing your specific recommendations as follows:
>
> - We have added an extended discussion about our assumptions on the cost function. In short, we chose positive definite costs because it greatly simplifies the statement of the theoretical results, and allows us to get our main points about sample efficiency across more clearly. We have added an extended discussion outlining how we believe the results in this paper provide a foundation for analyzing more complex costs in a series of future works.
>
> - We are currently running the ablation study on the quality of the value function at different training snapshots for the cartpole, and will upload plots of results soon. However, initial results indicate that 1) our approach is able to perform similarly to the baseline when our method has a poor value function (due to early termination) and the baseline has access to a good value function (no early termination) and 2) our method requires a higher discount factor and more samples when the lower quality value function is used instead of the final one.
>
> - For the quadruped tracking task, we are analyzing the possible sources for the tracking error and oscillations. Our current thoughts are:
>
>     - Leading the reference: During training we observed that at some iterations the policy led the target while at some iterations it trailed the target. We did not run the training to convergence, and stopped when we saw a satisfactory average tracking error. Thus, we believe that the fact that the final policy leads the reference can be explained by the random nature of the SAC algorithm.
>
>     - More oscillations: We believe two factors contribute to the oscillations of the learned policy: 1) RL policies are well known to produce oscillator policies on hardware, thus, this could be a feature of the RL algorithm (SAC) and not our method; and 2) the robot generally tends to oscillate more at higher velocities and our policy tends to achieve higher velocities than the nominal policy. The current objective we are using does not penalize oscillations, so adding a relevant feature to the costs might help decrease oscillations. We are currently investigating this.
>
> Please let us know if we have addressed your concerns and what additional changes to the paper you would like to see to improve your opinion.

---

> > ### Comment · Reviewer_XRvu · 2022-08-26
> > **Thank you for the responses**
> >
> > The authors have addressed most of my comments and I believe the paper has improved through the additional discussion, particularly regarding limitations/assumptions, as well as the new experimental evaluation. I did not see the plot(s) for bipedal walking and believe these could further strengthen the evaluation. I'll be happy to increase my score to weak accept.

---

> > > ### Author Response · Authors · 2022-08-28
> > > **Additional Content Added**
> > >
> > > We would like to thank the reviewer for their positive comments and additional feedback. We have now updated the draft of the paper to include the bipedal robot simulations, in which we show that we can fine-tune a model-based controller in face of significant dynamics mismatch with less than a minute of data, whereas the baseline takes 15 times as much data to reach a comparable average tracking performance. Furthermore, we have added a comparison with a baseline for the quadruped height-tracking additional experiments of our initial response.
> > >
> > > Following your advice, we have also added the quadrupedal velocity tracking experiments to our submitted video.
> > >
> > > Finally, we are attaching to this response a brief document in which we detail the additional comparison we have run on a simulator of the cartpole system to study the effects that using a poorer value function trained in simulation as CLF has on the performance of the later fine-tuning process. We will finalize this comparison and integrate it in the Appendix of the paper for the camera-ready submission.
> > >
> > > Finally, we wanted to ask the reviewer what they meant by "There are also several instances where the reader is referred to prior work (e.g. construction of CLF), while inclusion in the appendix would make the paper more self-contained." . Was it 1) to provide constructions for the specific CLFs used in this paper; or 2) a more general discussion of how to construct CLFs? We did not have time to address this during the rebuttal but will be happy to include it in the Appendix of the final draft.

---

### Author Response · Authors · 2022-08-28
**Current Draft + Updated Video**

Please find the current draft attached.

Link to video: https://drive.google.com/file/d/1C1p4flX3bGx13r345C80xgCqJnhNXUhj/view

---

### Meta-Review · Area_Chair_8Ama · 2022-08-13

**Recommendation:** Accept (Poster)
**Confidence:** 4

**Metareview:**

The paper introduces a theoretically-founded approach to an important problem of efficient fine-tuning. It is well-written, clear, and contains hardware experiments including a quadruped. However, the experimental evaluation is rather simple and lacks comparisons to baselines.

Main pros:
- The work is theoretically founded and analyzed
- The paper is well-written

Main cons:
- The limitations should be discussed in more detail
- The experimental evaluation is limited and does not fully demonstrate the contributions
- Lack of comparison with other papers that also use CLF term in the cost function of RL

Post-rebuttal update: The revised version of the paper largely addresses the main cons, and especially offered stronger experimental evaluation, further explanations and discussions.

**Best Paper Nomination:**

No

---

> ### Author Response · Authors · 2022-08-24
> **Overview of Changes**
>
> $\textbf{Please note: we have updated this response to reflect the current status of the paper draft and the new experiments we provide. }$
>
> Here we would like to briefly summarize the main changes to the paper we are working on in order to address the primary concerns of the reviewers:
>
> **Video**: We have attached an updated version of the supplementary video to include the velocity tracking hardware experiments. Due to time constraints, we were unable to include the video for the experiment with the unknown load, but will for the final submission.
>
> **Limitations**: We have extensively expanded the limitations section to discuss the main points raised by the reviewers.
>
> **Empirical Evaluation**:
>
> We have provided two new experimental tasks and have extensively performed baseline comparisons for all the results of the paper. In every case, our method outperforms the baselines by large margins.
>
> We are providing the following new experiment set-ups:
>
> - Quadruped gait tracking with unknown load ($\textbf{updated}$): We provide an additional hardware experiment where we learn a walking gait when an unknown load is strapped to the quadruped. We significantly decrease the tracking error to the target gait with only 1 minute of data. We performed a simulation benchmark comparison for this task, and demonstrated that the baseline approach took ~2x as much data to reach a similar of level of performances, and took significantly more data to consistently match the performance of our method.
>
> - Bipedal Walking  ($\textbf{updated}$): We applied our method to a simulated bipedal robot and have been able to obtain a stable walking gait in <1 minute of data in the face of significant model mis-match. We are able to accurately track the desired gait with less than 1 minute of simulated data. Meanwhile, the baseline fails to achieve the same level of performance even with ~15x as much data.
>
> - For these new experiments, please note that we did not have time to put all relevant details of the experiments in the appendix (network architectures, etc). We plan to include a section containing these details in the final submission.
>
> Baseline comparisons for initial experiments:
>
> - Cartpole: It appears that we did not adequately draw attention to the comparison between our approach and `standard’ fine-tuning approaches that we provided in Appendix D.1. of the initial submission. We maintain that this example clearly demonstrates that our approach improves the sample complexity of learning a stabilizing controller for this set up. We have moved this graph to the main paper to better highlight how we outperform the benchmark.
>
> - Quadruped Velocity Tracking: We are comparing our approach to standard methods using a simulator for the A1. In these experiments, our approach clearly outperforms the benchmark that uses a squared error cost function with SAC, as it can be seen in Figure 5 of the new version of the paper.
>
> - Inverted Pendulum: We have worked to clarify our description of the baselines we ran for this example
>
> - We have also benchmarked in simulation our approach in both of the additional experiments that we have provided in this new version of the draft (bipedal walking and quadruped gait tracking with unknown load) against typical squared error cost functions.
>
> $\textbf{Other papers with CLF term}$: We would like to thank Reviewer 3 for bringing the two new references to our attention and have thoroughly addressed these works in the new draft.  In short, we argue the novel Lyapunov-based actor-critic approaches from these references are $\textbf{algorithmic advances}$ which take advantage of Lyapunov theory in a principled fashion, while our approach is a $\textbf{design pipeline}$ aimed at reducing the sample complexity of learning a stable controller on hardware. In both references learning a CLF (typically from long-horizon rewards) is an important part of the algorithm. The contribution of these approaches, compared with standard actor-critic approaches, is to use a penalty approach to ensure the policy decreases the learned CLF. In contrast, our approach is to assume we already have a reasonable CLF candidate, and formulate a problem which can be greedily optimized to obtain a stabilizing controller (i.e. with a small discount factor). Thus, we feel our approach is actually complimentary to these works, as these approaches could be used to improve how we learn a `good’ CLF in simulation, as was done with the cartpole example.